# Similar but different: Profiling secondary school students based on their perceived motivational climate and psychological need-based experiences in physical education

**Gwen Weeldenburg**[1,2]*, **Lars B. Borghouts**[1], **Menno Slingerland**[1], **Steven Vos**[1,2]

1 School of Sport Studies, Fontys University of Applied Sciences, Eindhoven, The Netherlands,
2 Department of Industrial Design, Eindhoven University of Technology, Eindhoven, The Netherlands

* g.weeldenburg@fontys.nl

## Abstract

The purpose of this study was to provide more insight into how the physical education (PE) context can be better tailored to the diverse motivational demands of secondary school students. Therefore, we examined how different constructs of student motivation in the context of PE combine into distinct motivational profiles, aiming to unveil motivational similarities and differences between students' PE experiences. Participants were 2,562 Dutch secondary school students, aged 12–18, from 24 different schools. Students responded to questionnaires assessing their perception of psychological need satisfaction and frustration, and perceived mastery and performance climate in PE. In order to interpret the emerging profiles additional variables were assessed (i.e. demographic, motivational and PE-related variables). Two-step cluster analysis identified three meaningful profiles labelled as *negative perceivers*, *moderate perceivers* and *positive perceivers*. These three profiles differed significantly with regard to perceived psychological need satisfaction and frustration and their perception of the motivational climate. This study demonstrates that students can be grouped in distinct profiles based on their perceptions of the motivational PE environment. Consequently, the insights obtained could assist PE teachers in designing instructional strategies that target students' differential motivational needs.

## Introduction

One of the main aims of Physical Education (PE) in the Netherlands is to provide students with competencies that enable and encourage them to participate in sports and physical activity (PA) in and outside of the school setting [1,2]. In order for students to develop these competencies within the psychomotor, cognitive and affective domain, they should be sufficiently motivated to actively partake in the PE lessons. Unfortunately, motivation of Dutch students has been found to decrease from the end of primary school into secondary school [3]. The PE teacher is therefore challenged to find ways to motivate all students to actively engage in physical activities during PE lessons.

**Funding:** The authors received no specific funding for this work.

**Competing interests:** The authors have declared that no competing interests exist.

Previous research has shown that by managing the learning environment in such a way that it meets the motivational demands of students, the PE teacher is able to positively influence the engagement in and attitude towards PA within PE [4–8]. However, satisfying the motivational needs of every student is a complex task given the substantial heterogeneity present in PE-classes. It is up to the PE-teacher to find ways to cater to the unique and differential needs of all students, not only regarding psychomotor skills but also to affective differences. Students differ in the ways they can be motivated for PE [6,9,10–14]. However, deliberately and systematically creating and managing a differentiated motivational learning environment is by no means self-evident for PE teachers in the Netherlands.

In order to adequately address students' needs in PE and to better tailor the PE-context, determining the variety in motivational demands of secondary school students is relevant. The purpose of this paper is therefore to gain more insight into students' perception of the motivational learning climate in secondary school PE in the Netherlands. We explore whether different constructs of student motivation in the context of PE (i.e. perceived psychological need satisfaction and need frustration, and perceived teacher-initiated mastery climate versus performance climate) combine into distinct student profiles, aiming to unveil motivational similarities and differences between students' experiences in PE.

## Motivation

Student motivation toward PE is a complex and dynamic process [11]. In order to understand and eventually influence this process, multiple perspectives should be considered. In the context of PE two major theoretical frameworks of motivation can be discerned: achievement goal theory (AGT) [15–17] and self-determination theory (SDT) [18]. Both theories stress the importance of the learning environment created by the teacher [19].

## Achievement Goal Theory

According to AGT individuals strive to perceive themselves as competent [20]. In doing so, different achievement goals can be discerned which differentially guide and influence the behaviour, cognition and emotion [21] and how an individual defines competence and success [11,19]. According to the 2 x 2 achievement goal framework [17,22] four different achievement goals are distinguished. Goals that focus on mastering the requirements of a task and on self-improvement, are defined as *mastery approach goals* (MAp). *Mastery avoidance goals* (MAv) are aimed at the avoidance of task-defined or self-defined failure [17]. Goals in which the individual demonstrates his or her ability through outperforming others are defined as *performance approach goals* (PAp), whereas *performance avoidance goals* (PAv) refer to avoiding performing worse than others [23].

In addition to the four achievement goals, AGT literature distinguishes between two motivational climates: *mastery* (i.e., task-involving) *climate* and *performance* (i.e., ego-involving) *climate* [24–26]. A mastery climate is characterized as an environment in which students perceive they are rewarded for effort, personal development, learning, cooperation and individual improvement, whereas a performance climate promotes and facilitates social comparison between students by rewarding superior performance [5]. Overall, perceived mastery climates are positively associated with several adaptive constructs, such as higher levels of self-determined regulation, greater satisfaction of the basic psychological needs and greater enjoyment [27–29], while perceptions of a performance climate have been linked to less desirable outcomes, such as boredom, anxiety and antisocial moral attitudes [27,30].

By creating a particular motivational climate, the PE teacher may influence achievement goals of students. However, students' reasons for pursuing each of the achievement goals can

vary [31]. It is therefore desirable to separate the aims or targets of the goals ('what') from the underlying reasons or goal motivation ('why') [21,23]. A framework that is eminently suitable for studying why people display certain behaviours is the self-determination theory.

**Self-determination theory.** Self-determination theory (SDT) conceptualizes students' *autonomous motivation* as pursuing goals because they find them interesting, enjoyable or exciting (i.e., intrinsic motivation), because the goal is in alignment with the values and norms of the student (i.e., integrated motivation), or because the student perceives personal relevance (i.e., identified motivation). In contrast, pursuing goals in order to avoid feelings of guilt (introjected motivation), or in order to obtain rewards or avoid negative consequence (external motivation) is referred to as *controlled motivation* [32]. Finally, when intentionality and energy or desire to act is missing, due to lack of concern or valuation of the activity, or the lack of perceived competence or positive efficacy beliefs, this is referred to as amotivation [6,33].

Previous studies in the context of PE showed that autonomous forms of motivation (i.e., intrinsic motivation, integrated motivation, identified motivation) can be associated with more adaptive cognitive, affective and behavioural outcomes than controlled motivation (Aelterman et al., 2016). Autonomously motivated students show more PA [6,34,35], concentration [36], effort, engagement and persistence [37] during PE lessons. In contrast, students who feel internally or externally pressured (controlled motivation) show less adaptive outcomes during PE classes, such as lower engagement [34], boredom, and unhappiness [38,39]. Amotivation relates negatively to effort [40], rated engagement [34], and well-being [38].

The prerequisite for autonomous motivation is the satisfaction of three basic psychological needs: autonomy, competence and relatedness [18,41]. *Autonomy* refers to regulation by the self. *Competence* refers to the need to experience some level of effectiveness and confidence. The concept of *relatedness* refers to the need to feel connected with others, to feel included and cared for by others. When these needs are satisfied students are more likely to be driven by autonomous forms of motivation. Whereas the frustration of these needs will result in the feelings of pressure and being forced to participate in PE (i.e., autonomy frustration), the feeling of failure and being incompetent to deal effectively with the situation (i.e., competence frustration) and feelings of loneliness and being disconnected from others (i.e., relatedness frustration). In these circumstances, the student will more likely participate in PE on basis of more controlling motives [42,43]. Need satisfaction and need frustration must be considered as separate constructs and do not necessarily fall along a single continuum as opposites [42,44]. In other words, high need satisfaction does not automatically imply low need frustration.

The (social) learning environment created by the PE teacher (i.e., motivational climate) [45] has the potential to influence student motivational regulations for participating in PE by fostering the experiences of need satisfaction and prevent need frustration during PE lessons [46,36,47]. Previous studies showed that need-supportive teaching behaviour relates positively to autonomous (high-quality) forms of motivation for PE (e.g. [48]) and to subsequent positive outcomes, including enjoyment in PE (e.g. [49]). In contrast, need thwarting teacher behaviours are related to controlled motivation and maladaptive student outcomes [44]. In order to optimise the match between the learning environment and the motivational demands of students, the similarities and differences between students should be taken into account.

**Motivational profiling.** In motivational profiling, separate motivational dimensions are organized into profiles representing naturally occurring combinations of these dimensions within subgroups of individuals [31]. By identifying the common characteristics of the profiles, this enables tailoring interventions to the different needs and wants of specific groups [50]. In the context of PE, this approach of identifying different groups may allow for the identification of groups of students that need extra guidance or attention. Profiling could increase the efficiency and effectiveness of teacher interventions and enhance students' engagement across

profiles by aligning teaching style, curricular and pedagogical decisions, and learning tasks and strategies with the motivational demands of students in a specific profile. For example, it is conceivable that students in one profile benefit more from providing choice during learning activities than students in another profile. Or students in one profile may thrive better in a performance orientated learning climate than students in another profile. Past studies mainly focused on profiling based on type of motivation (e.g. [6], type of motivation regulation [13], or type of achievement goal orientation [51]. The present study aims to expand on the current profiling research, by exploring how students perceive the learning environment in PE by combining the motivational frameworks of AGT and SDT. In line with the study conducted by Vansteenkiste and Mouratidis [31], and several other studies [20,43,52] we hypothesise that profiling students based on a combination of theoretical motivational frameworks, can generate new useful insights for professional practice.

PE teachers can directly impact the motivational climate by differentially emphasizing a mastery or performance climate as well as the perceived need satisfaction and frustration of their students. This study therefore investigates how these constructs combine into distinct motivational profiles in secondary school PE. In order to subsequently characterize, validate and compare these profiles, we explore whether profile membership is related to demographic variables, motivational regulation, achievement goal orientation, valuation of PE and self-reported leisure time sport participation (LTSP).

## Methods

### Participants and procedure

Participants were 2,562 secondary school students (44% boys; 56% girls), aged 12–18 years ($M$ age = 14.65; $SD$ = 1.38), across 24 different secondary schools from 16 different cities in the Netherlands. PE in the Netherlands is mixed gender grouped and mandatory within all types of secondary school education. On average, secondary school students in the Netherlands participate in two lessons (of 50–60 minutes) of compulsory PE per week throughout the school year [53].

Ethical approval was obtained by the Ethical Research Committee of Fontys University. Permission to collect data with the students was obtained from the local school boards. The students were explained that the participation in the study was voluntary, there was guarantee of confidentiality and anonymity and that non-participation would not cause them any harm. They could also choose to withdraw from the study at any time without giving any reason.

Secondary schools were recruited from the existing network of the university. The questionnaire was administered to students at school near the end of the school year. The questions concerned students' perceptions of the PE motivational climate and experiences during the past schoolyear. The students took an average of 10 minutes to complete the questionnaire and were supervised by a teacher (i.e. not the PE-teacher) who was well informed about the procedure.

### Measures

**Motivational variables used to construct profiles.**

**Psychological need satisfaction and frustration.**   Students' perceived psychological need (autonomy, competence and relatedness) satisfaction and frustration during the PE-lessons in general, were assessed by a modified version of the validated Basic Psychological Need Satisfaction and Frustration Scale (BPNSFS) [54]. The total number of items in the questionnaire (i.e. 24) used in this research was reduced to enhance the usability for the target group. For each construct, experts (N = 4) independently determined which three of four items per construct

**Table 1. Overview and descriptive statistics of motivational constructs used for profiles.**

| Motivational construct | Items | α | n | Mean | Min | Max | SD |
|---|---|---|---|---|---|---|---|
| Autonomy satisfaction | 3 | .75 | 2,562 | 3.17 | 3.01 | 3.27 | .80 |
| Autonomy frustration | 3 | .77 | 2,562 | 2.49 | 2.27 | 2.76 | .91 |
| Competence satisfaction | 3 | .83 | 2,562 | 3.44 | 3.28 | 3.52 | .81 |
| Competence frustration | 3 | .84 | 2,562 | 2.24 | 2.19 | 2.30 | .92 |
| Relatedness satisfaction | 3 | .75 | 2,562 | 3.78 | 3.68 | 3.93 | .79 |
| Relatedness frustration | 3 | .70 | 2,562 | 1.90 | 1.79 | 2.12 | .72 |
| Mastery climate | 6 | .87 | 2,562 | 3.84 | 3.65 | 4.13 | .70 |
| Performance climate | 6 | .79 | 2,562 | 2.13 | 1.87 | 2.35 | .70 |

represented the construct best, and then conferred to reach consensus on which items to retain. In the resulting questionnaire, the stem *'In general during PE-lessons. . .'* was followed by three items reflecting autonomy satisfaction (e.g., 'I feel a sense of choice and freedom in the things I undertake'), autonomy frustration (e.g., 'I feel forced to do many things I wouldn't choose to do'), competence satisfaction (e.g., 'I feel confident that I can do things well'), competence frustration (e.g., 'I have serious doubts about whether I can do things well'), relatedness satisfaction (e.g., 'I feel close and connected with other people who are important to me') and relatedness frustration (e.g., 'I feel excluded from the group I want to belong'). The item-set was tested for usability and language suitability in a pilot study (N = 10). All items were rated on a 5-point Likert scale ranging from 1 ('not at all true for me') to 5 ('very true for me'). Cronbach's Alpha scores were calculated for each construct (see Table 1).

**Perceived mastery and performance climate.** Students' perceived teacher-initiated motivational climate during PE-lessons throughout the whole school year was measured by a modified version of the Motivational Climate Scale for Youth Sports (MCSYS) [55]. The item-set was tested for usability and language suitability in a pilot study (N = 10). Students responded to the items on a 5-point Likert scale ranging from 1 ('not at all true for me') to 5 ('very true for me'). Total scores for each scale were computed by averaging across the items. The MCSYS is based on the content of the Perceived Motivational Climate in Sport-2 (PMCSQ-2) [56] and wording was slightly adapted to the PE context. It consists of six items indexing a mastery climate (e.g., 'The PE teacher told us that trying our best was the most important thing') and six items assessing a performance climate (e.g., 'The PE teacher told us to try to be better than our teammates'). The MCSYS is age-appropriate for the present sample and has demonstrated good internal consistency for each subscale (> .70) and adequate test-retest reliability (.84 for mastery, .76 for ego) [55].

The internal consistencies of the modified questionnaire were satisfactory with Cronbach's Alpha's of .87, and .79 for subscale mastery (task-involving) climate and performance (ego-involving) climate respectively (see Table 1).

**Constructs and variables used to describe profiles.** To interpret, describe and compare the emerging profiles, demographic variables, motivational variables and PE-related variables were used. The demographic variables consisted of gender and age. The motivational variables were achievement goal orientation and motivational regulation. The PE-related variables concerned students' general valuation of PE and leisure time sport participation (LTSP).

**Achievement Goal Orientation.** The validated 2x2 Achievement Goal in Physical Education Questionnaire (2x2 AGPEQ) [14] was used to measure the achievement goal orientation of students in the context of PE. The 12-item scale reflects the four achievement goals (3 items each): mastery-approach goal (e.g., 'I want to learn as much as possible from PE class'), mastery-avoidance goal (e.g., 'I worry that I may not learn all that I possibly could in PE class'),

performance-approach goal (e.g., 'It is important for me to do better than other students in this PE class'), and performance-avoidance goal (e.g., 'I just want to avoid doing poorly in PE class'). A Likert scale was used ranging from 1 ('not at all true for me') to 5 ('very true for me'). Internal consistencies of mastery approach goal, mastery avoidance goal and for performance approach goal were satisfactory (see Table 2). The Cronbach's Alpha of .56 for performance avoidance goal was less satisfactory. However, Kline [57] notes that when dealing with psychological constructs, values below .70 can, realistically, be expected. Furthermore, the value depends on the number of items on the scale [58], and the number of items within the scale of performance avoidance goal is only three. Given that the α value was in line with the original findings of the validation research of Wang et al. [14] and the fact that deleting items from the analysis had no relevant positive effect on the Cronbach's Alpha of the scale, it was decided to keep the performance avoidance goal scale for further analysis.

**Motivational regulation.**  Students' motivational regulations towards PE in general, were assessed by using an adapted version of the Behavioural Regulations in Physical Education Questionnaire (BRPEQ) [34]. The original questionnaire (i.e. BRPEQ) includes 20 items reflecting autonomous motivation (i.e., intrinsic motivation (4 items); identified regulation (4 items)), controlled motivation (i.e., introjected regulation (4 items); external regulation (4 items)), and amotivation (4 items). Given the usability and feasibility of the total questionnaire applied in this study, the number of items of the BRPEQ was reduced from 20 to 12 items on the basis of the factor analyses by Aelterman et al. [34] and Haerens et al. [42], and after personal communication with the authors of these studies. The introductory stem '*In general I put effort in PE class. . .*' was followed by 4 items reflecting autonomous motivation (2 x 2 items; e.g., 'because I enjoy it'; 'because I find PE personally meaningful'), 4 items reflected controlled motivation (2 x 2 items; e.g., 'because I have to prove myself'; 'because I feel the pressure of others to participate') and 4 items reflected amotivation (e.g., 'I find PE a waste of time'). All items were rated on a 5-point Likert scale ranging from 1 ('not at all true for me') to 5 ('very true for me'). Subscale scores (i.e., higher order factors) were calculated by averaging the four items. Cronbach's Alpha scores were calculated for each subscale (see Table 2).

**Valuation of PE.**  Students' general valuation of PE was measured using a single item ('What grade would you give PE class in general?'). This item was scored on a 10-point scale (1 = very poor; 10 = very good). This satisfaction measurement is familiar to students as it is part of the Dutch education and evaluation system.

**Leisure Time Sport Participation (LTSP).**  The Netherlands have a strong system and tradition of sports clubs. With 62 sports clubs on average per municipality, sports clubs play a central role in the Dutch sports system. The self-reported amount of students' sport participation during leisure time was measured using the single item '*Do you practice sports/ sports activities outside the school setting*? (*for example, football, field hockey, tennis, fitness, swimming,*

**Table 2. Overview and descriptive statistics of motivational constructs to describe and compare profiles.**

| Motivational construct | Items | α | n | Mean | Min | Max | SD |
|---|---|---|---|---|---|---|---|
| Performance Approach Goals | 3 | .83 | 2,562 | 2.59 | 2.36 | 2.91 | .95 |
| Performance Avoidance Goals | 3 | .56 | 2,562 | 2.85 | 1.99 | 3.51 | .82 |
| Mastery Approach Goals | 3 | .80 | 2,562 | 3.52 | 3.38 | 3.72 | .82 |
| Mastery Avoidance Goals | 3 | .80 | 2,562 | 2.34 | 2.18 | 2.56 | .89 |
| Autonomous Motivation | 4 | .88 | 2,562 | 3.49 | 3.25 | 3.72 | .94 |
| Controlled Motivation | 4 | .68 | 2,562 | 2.04 | 1.72 | 2.36 | .70 |
| Amotivation | 4 | .91 | 2,562 | 2.16 | 2,13 | 2.22 | 1.02 |

*dancing, gymnastics, etc.)*'. The response options were (1) 'Less than 1 time a week on average'; (2) 1 or 2 times a week on average; (3) '3 or more times a week on average'.

## Data analysis

An exploratory cluster analysis was used to group students into homogeneous clusters representing similar perceptions of the motivational PE learning climate. Cluster analysis is a convenient method for identifying groups of individual students that are similar to each other but different from individuals in other groups [59]. The clustering variables in this study were autonomy satisfaction, autonomy frustration, competence satisfaction, competence frustration, relatedness satisfaction, and relatedness frustration based on the theoretical framework of the self-determination theory, together with the variables mastery climate and performance climate based on the achievement goal theory.

The two-step cluster analysis procedure using the log-likelihood measure was conducted. This procedure automatically clusters similar groups of respondents within data sets [60]. The first step of the two-step procedure is assigning the original cases to pre-clusters based on distance measure in order to reduce the size of the matrix that contains distances between all possible pairs of cases. In the second step, the standard hierarchical clustering algorithm was used on the pre-clusters resulting from the first step, to define the optimal number of clusters. The Schwarz Bayesian information criterion (BIC) was used to determine the optimal number of clusters. The BIC is considered as an objective and useful selection criterion [61,62,63].

After the first cluster solution was formed measures of validity were applied. First, following recommendations of Norusis [60] and Dietrich, Rundle-Thiele & Kubacki [64], the silhouette measure of cohesion and separation was conducted to measure the validity of the within and between cluster distances. Second, a multivariate analysis of variance (MANOVA) was performed to assess latent differences among clusters. Followed-up by analysis of variances (ANOVA) on each cluster variable to assess significant differences among cluster solutions. Subsequently, the input (predictor) importance was measured to determine the importance of variables in a cluster. Third, the data-set was randomly split into two (using 'select cases' option in SPSS) and another two-step cluster analysis was performed to provide the last validation measure.

To present the characteristics of the final clusters descriptive statistics were carried out. Each profile was then labelled based on the motivational variable scores and with which the students in that profile were characterized. Chi-squared tests ($p < 0.05$ was considered significant) with post hoc testing (through z-scores and adjusted p-values; Bonferroni-method), and one-way ANOVA were used to examine the possible link between profile membership and demographic characteristics (i.e., gender, age), motivational characteristics (i.e., achievement goal orientation, motivational regulation), and PE-related characteristics (i.e., leisure time sport participation and valuation of PE).

## Results

In total 2,562 students were included, whilst data from 268 students was deleted because of invalid, partially completed questionnaires. The two-step cluster analysis identified three meaningful clusters based on Schwarz's BIC and the highest Log-likelihood distance measures. In clusters 1, 2, and 3, there were 834 (32.5%), 1,080 (42.2%), and 648 (25.3%) students, respectively. In Fig 1, the three clusters and their results in the clustering variables are presented. This three-cluster solution was deemed acceptable based on the silhouette measure value of 0.3. The variable importance rating was 0.58 or higher for all variables (i.e., autonomy satisfaction = 1.00; competence frustration = 0.91; relatedness frustration = 0.83; competence

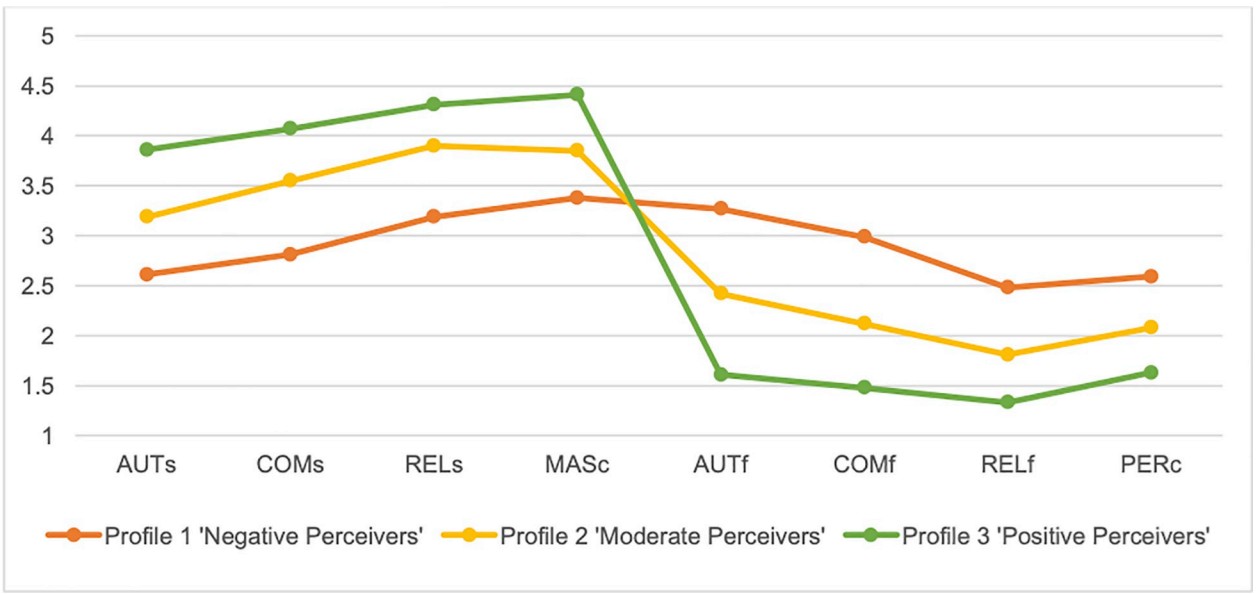

**Fig 1. Graphic display of profiles based on eight motivational factors.** AUTs = Autonomy satisfaction; COMs = Competence satisfaction; RELs = Relatedness satisfaction; MASc = Mastery Climate; AUTf = Autonomy frustration; COMf = Competence frustration; RELf = Relatedness frustration; PERc = Performance Climate.

satisfaction = 0.80; autonomy satisfaction = 0.77; relatedness satisfaction = 0.67; mastery climate = 0.66; performance climate = 0.58). In the halved sample validation, a two-step cluster analysis also produced a final cluster solution with three similar/identical profiles (silhouette measure value = 0.3) and characteristics.

Based on MANOVA, clustering variables were significantly different between clusters (Pillai's Trace = .82, F (16, 5106) = 222.93, p < .001). Follow-up ANOVA's with Games-Howell post-hoc tests revealed significant differences between all three profiles on each of the eight clustering variables (see Table 3), supporting the three-cluster solution.

Based on the scores on the clustering variables, the first profile 1 was labelled *negative perceivers*, the second profile was labelled *moderate perceivers*, and the third profile was labelled *positive perceivers*. The labels are based on the relative differences between the clusters rather

**Table 3. Descriptive statistics for the three profiles emerged from two-step cluster analysis.**

| | Overall | | Profile 1 negative perceivers (*n* = 834) | | Profile 2 moderate perceivers (*n* = 1080) | | Profile 3 positive perceivers (*n* = 648) | | | | |
|---|---|---|---|---|---|---|---|---|---|---|---|
| | **M** | **SD** | **M** | **SD** | **M** | **SD** | **M** | **SD** | *p* | *F* | *ω* |
| Autonomy satisfaction | **3.17** | **.77** | 2.61[a] | .77 | 3.19[b] | .56 | 3.86[c] | .61 | p < .001[a-b, a-c, b-c] | 678.22 | .59 |
| Autonomy frustration | **2.49** | **.83** | 3.27[a] | .83 | 2.42[b] | .58 | 1.61[c] | .52 | p < .001[a-b, a-c, b-c] | 1172.56 | .69 |
| Competence satisfaction | **3.44** | **.80** | 2.81[a] | .80 | 3.55[b] | .55 | 4.07[c] | .59 | p < .001[a-b, a-c, b-c] | 712.73 | .60 |
| Competence frustration | **2.24** | **.96** | 2.99[a] | .96 | 2.12[b] | .59 | 1.48[c] | .52 | p < .001[a-b, a-c, b-c] | 833.45 | .63 |
| Relatedness satisfaction | **3.78** | **.84** | 3.19[a] | .84 | 3.90[b] | .52 | 4.31[c] | .57 | p < .001[a-b, a-c, b-c] | 572.85 | .56 |
| Relatedness frustration | **1.90** | **.78** | 2.48[a] | .78 | 1.81[b] | .46 | 1.33[c] | .41 | p < .001[a-b, a-c, b-c] | 744.28 | .61 |
| Mastery climate | **3.84** | **.73** | 3.38[a] | .73 | 3.85[b] | .53 | 4.41[c] | .45 | p < .001[a-b, a-c, b-c] | 561.89 | .55 |
| Performance climate | **2.13** | **.74** | 2.59[a] | .74 | 2.08[b] | .51 | 1.63[c] | .49 | p < .001[a-b, a-c, b-c] | 484.02 | .52 |

**Note**. Means and SD's were on a 5-point Likert scale. [a-b, a-c, b-c] indicate significant differences between profiles based on Games-Howell post-hoc tests.

**Table 4. Group differences on demographic characteristics.**

| | Total | Profile 1 negative perceivers | Profile 2 moderate perceivers | Profile 3 positive perceivers | | | |
|---|---|---|---|---|---|---|---|
| *Demographic characteristics* | | N (%) | N (%) | N (%) | *p* | | |
| Gender | | | | | | | |
| *Male* | | 364 (32%) | 490 (43%) | 284 (25%) | p>.05 | | |
| *Female* | | 470 (33%) | 590 (41%) | 364 (26%) | | | |
| | *M (SD)* | *M (SD)* | *M (SD)* | *M (SD)* | *p* | *F* | *ω* |
| Average age | **14.65 (1.38)** | 14.64 (1.39) | 14.62 (1.39) | 14.73 (1.37) | p>.05 | 1.32 | .02 |

Note. Percentages are row percentages

than on absolute values within the three cluster variables. All profiles differ significantly (p<0.01) with regard to autonomy satisfaction, autonomy frustration, competence satisfaction, competence frustration, relatedness satisfaction, relatedness frustration, and perception of the motivational climate (mastery versus performance).

*Moderate perceivers* is the largest of the three profiles (N = 1,080; 42.2%). Students in this profile report higher levels of perceived need satisfaction than need frustration, and a perceived mastery climate rather than a performance climate. Their scores approximately correspond to the mean of the entire population. *Negative perceivers* (N = 834; 32.5%) report the lowest perceived satisfaction and highest perceived frustration of all three basic psychological needs. Also, they score lowest on perceived mastery climate and highest on perceived performance climate, even though the former still scores higher than the latter. *Positive perceivers* (N = 648; 25.3%) reported the highest and lowest levels of psychological need satisfaction and frustration, respectively. They perceive the PE class climate as highly mastery oriented.

To further describe the profiles, we compared additional characteristics of students in the three profiles, demographic characteristics (i.e., gender, age), motivational characteristics (i.e., achievement goal orientation, motivational regulation), and PE-related characteristics (leisure time sport participation, valuation of PE). Chi-squared tests and one-way ANOVA's were used to examine the possible link between profile membership and these additional characteristics of students.

No significant profile assignment effect by gender ($\chi^2$ (2, n = 2,562) = .69, p >.05), and average age ($F$(2, 2559) = 1.322, p>.05) was found. Males and females were equally distributed across the three profiles, and no significant age differences between groups were found (see Table 4).

Analysis of variance with post-hoc tests revealed a significant link between cluster membership and students' achievement goal orientation and motivational regulation (see Table 5). Concerning the achievement goal orientation, students in all three profiles report the highest scores on mastery approach goals (see Table 5). However, the score of the *negative perceivers* is significantly lower (3.13/5) than the *moderate* (3.5/5) and *positive perceivers* (4.02/5), and almost equal to their score on the performance avoidance goals (3.08/5). The *positive perceivers* showed the clearest contrast between mastery approach goals (4.02/5) and mastery avoidance goals (1.88/5). With respect to the motivational regulation the *positive perceivers* showed significantly higher levels of autonomous motivation than both *moderate* perceivers and *negative perceivers*. *Negative perceivers* showed significantly higher levels of controlled motivation and amotivation compared to *moderate perceivers* and *positive perceivers* (see Table 5).

For leisure time sport participation (LTSP), significant differences were also found between profiles, $\chi^2$ (4, n = 2562) = 74.33, p < .001. Students who reported LTSP less than once a week

**Table 5. Group differences on motivational characteristics.**

| | Total | Profile *negative perceivers* | Profile *moderate perceivers* | Profile *positive perceivers* | | | |
|---|---|---|---|---|---|---|---|
| *Motivational characteristics* | *M (SD)* | *M (SD)* | *M (SD)* | *M (SD)* | *p* | *F* | *ω* |
| Achievement goal orientation | | | | | | | |
| *Mastery Approach Goal* | 3.52 (.82) | 3.13[a] (.86) | 3.5[b] (.69) | 4.02[c] (.70) | p < .01[a-b, a-c, b-c] | 253.05 | .41 |
| *Mastery Avoidance Goal* | 2.34 (.90) | 2.76[a] (.99) | 2.29[b] (.74) | 1.88[c] (.74) | p < .01[a-b, a-c, b-c] | 206.10 | .37 |
| *Performance Approach Goal* | 2.59 (.95) | 2.51[a] (.94) | 2.61[b] (.88) | 2.65[c] (1.06) | p < .05[a-c] | 4.14 | .05 |
| *Performance Avoidance Goal* | 2.85 (.92) | 3.08[a] (.92) | 2.81[b] (.70) | 2.61[c] (.79) | p < .01[a-b, a-c, b-c] | 66.40 | .22 |
| Motivational regulation | | | | | | | |
| *Autonomous motivation* | 3.49 (.94) | 2.88[a] (.95) | 3.52[b] (.72) | 4.20[c] (.70) | p < .01[a-b, a-c, b-c] | 495.914 | .53 |
| *Controlled motivation* | 2.04 (.70) | 2.36[a] (.78) | 2.00[b] (.55) | 1.70[c] (.62) | p < .01[a-b, a-c, b-c] | 193.813 | .36 |
| *Amotivation* | 2.16 (1.02) | 2.90[a] (1.08) | 2.01[b] (.76) | 1.45[c] (.63) | p < .01[a-b, a-c, b-c] | 555.731 | .55 |

Note. Means and SD's were on a 5-point Likert scale. [a-b, a-c, b-c] indicate significant differences between profiles based on Games-Howell post-hoc tests.

on average, were overrepresented in the cluster *negative perceivers* (44%) and underrepresented in the cluster *positive perceivers* (16.3%). Of the students who reported LTSP for three times or more a week on average, there were 26.4% *negative perceivers*, 43.3% *moderate perceivers* and 30.2% *positive perceivers* (see Table 6). These active students were significantly underrepresented in the cluster *negative perceivers* and overrepresented in the cluster *positive perceivers*. Also, a significant link between cluster membership and students' valuation of PE in general was discovered (see Table 6). *Negative perceivers* value PE significantly less (M = 6.34/10) than *moderate perceivers* (M = 7.40/10), and *positive perceivers* (M = 8.22/10).

## Discussion and conclusions

To gain more in-depth knowledge of how the PE-context can be tailored to the differential motivational demands of secondary school students in the Netherlands, the present study aimed to define motivational profiles of secondary school students based on their levels of perceived psychological need satisfaction and need frustration, as well as perceived teacher-initiated mastery climate versus performance climate during PE-lessons.

Cluster analysis revealed three motivational profiles which differ gradually from each other. *Negative perceivers* report relatively high levels of psychological need frustration and relatively low levels of psychological need satisfaction during PE. These students perceive feelings of pressure and feel more insecure about their competence in PE. Although *negative perceivers* score somewhat higher on relatedness satisfaction than frustration, they are not outspokenly

**Table 6. Group differences on PE-related characteristics.**

| | Total | Profile 1 *negative perceivers* | Profile 2 *moderate perceivers* | Profile 3 *positive perceivers* | | | |
|---|---|---|---|---|---|---|---|
| *PE-related characteristics* | *N* | *N (%)* | *N (%)* | *N (%)* | *p* | | |
| LTSP | | | | | | | |
| *<1/week in average* | 466 | 204 (44%) | 186 (40%) | 76 (16%) | p < .01 | | |
| *1-2/week in average* | 674 | 254 (38%) | 278 (41%) | 142 (21%) | | | |
| *≥3/week in average* | 1422 | 376 (26%) | 616 (43%) | 430 (30%) | | | |
| | *M (SD)* | *M (SD)* | *M (SD)* | *M (SD)* | *p* | *F* | *ω* |
| Valuation of PE | 7.26 (1.59) | 6.34[a] (1.94) | 7.4[b] (1.16) | 8.22[c] (.94) | p < .01[a-b, a-c, b-c] | 328.79 | .45 |

Note. Percentages are row percentages. [a-b, a-c, b-c] indicate significant differences between profiles based on Games-Howell post-hoc tests.

positive about the bond they have with peers either. In addition, these students report the lowest levels of perceived mastery-orientated learning climate of the three profiles. However, they still perceive the climate as more mastery-orientated than performance-orientated. The *negative perceivers* seem to be driven by both mastery *approach* and performance *avoidance* goals. Even though most students in this profile participate frequently in sports activities outside school, they are the least active. Students in this profile value PE distinctly lower than the students in the other two profiles. In contrast, the *positive perceivers* report the lowest levels of need frustration and the highest levels of need satisfaction. They report strong feelings of autonomy, activities undertaken in PE are perceived as interesting, and these students feel highly competent to achieve PE-goals. The *positive perceivers* feel strongly connected with peers and mastery approach goals strongly drive the students in this profile. They show the most explicit contrast between the mastery approach and mastery avoidance goals. Students in this profile value PE very positively. The *moderate perceivers* generally show scores that are close to the mean of the total population. In contrast to the negative perceivers, these students report higher levels of support than frustration for all psychological needs. Nevertheless, they do not perceive particularly strong feelings of autonomy support. The students in this profile perceive the PE learning climate as mastery-oriented and seem to be driven by mastery approach goals in particular.

Previous studies in the context of PE have demonstrated that perceived psychological need satisfaction correlates positively to autonomous forms of motivation, while perceived need frustration is associated with controlled forms of motivation and amotivation [42,65,66,67]. In line with these findings and the tenets of SDT, students in the *negative perceivers'* cluster show higher levels of controlled motivation and amotivation, and lower levels of autonomous motivation compared to the other clusters, while *positive perceivers* report high levels of autonomous motivation and low levels of controlled motivation and amotivation.

Compared to the other groups, *negative perceivers* experience the highest levels of relatedness frustration. Nevertheless, these students still perceive more feelings of relatedness satisfaction than frustration. This could indicate that, despite having a more negative experience during PE in terms of need frustration, these students do not feel excluded from the class nor being disliked by significant peers. Hence, we argue that interventions to improve motivational climate for these students could be focused more on supporting feelings of competence and autonomy, rather than relatedness. This line of argument is supported by research [40,68] which argues that some psychological needs (specifically perceived competence) are more influential predictors of autonomous forms of motivations than others.

Previous studies (for an overview see [27]) consistently indicate that a mastery climate is associated with a range of highly adaptive outcomes, including autonomous forms of motivational regulations and high levels psychological need satisfaction. A mastery (or task-involving) climate is thought to enhance the perception of autonomy and competence support, because achievement is based on self-referenced criteria and is therefore more self-controlled. In such a climate, feelings of success are more readily achievable than in a performance (or ego-involving) climate, with normative-based criteria [69,70]. A performance climate in PE has recently been found to positively relate to basic psychological need frustration, amotivation and boredom [71]. Our study endorses these findings. Moderate perceivers and positive perceivers experience a mastery PE climate and concurrently score relatively high on psychological need satisfaction and autonomous motivation. Although negative perceivers still perceive the PE climate as more mastery than performance oriented, they score only slightly above scale average on the former. Compared to the other profiles, they report relatively high levels of perceived competence- and autonomy frustration, controlled motivation and amotivation. Jaakkola et al. [72] argue that for some students the self-referenced learning climate might well

be more important than their perception of psychological need satisfaction. Our study suggests that particularly for the group of negative perceivers, instructional and pedagogical strategies should be aimed at improving their level of perceived mastery climate.

Compared to the other two profiles, negative perceivers report the highest levels of achievement goal avoidance, in particular the performance avoidance goals. This implies that negative perceivers have the tendency to withdraw themselves from situations during the PE lesson where comparison with peers is prominent, such as competitive game situations or when being assessed. Previous research indicates that this achievement goal avoidance predicts maladaptive outcomes such as lower satisfaction, self-esteem, self-confidence, performance and vitality [20,43,73]. Various studies have suggested (e.g. [74]) that these achievement goal constructs can be positively influenced by creating a high mastery/low performance climate. For example, by having students focus on a specific task performance within a game situation or by working in same skill level groups, in which social comparison is avoided as much as possible. However, in order to determine if students in a given profile may thrive better in a specific learning climate (i.e. performance or mastery climate) or benefit more from, for example, providing choices during learning tasks (i.e. autonomy support) than students in another profile, further intervention studies would be needed.

In the present study, we found no significant differences between the three motivational profiles in terms of gender and age. Concerning gender, this finding is consistent with profiling studies by Boiché et al. [10] and Ntoumanis [75] who also found no association between students' gender and membership of the favourable or less favourable motivational profiles. However, other studies have reported conflicting findings, reporting girls to be overrepresented in either the more [76] or less desirable profiles [62,77]. These contrasting findings may stem from differing methodological approaches but could also represent a cultural or educational difference between the contexts from which the groups of students were sampled. Regardless, from a gender equality perspective, we believe our findings are favourable in that the PE teachers in the schools of our sample apparently succeed in creating learning environments that neither favour boys nor girls in the less desirable profiles. Our findings regarding age contrast with research indicating that older students are overrepresented in low-quality motivation clusters, and younger students being underrepresented [10,62,76].

## Limitations

Since our study employed a cross-sectional design, causality cannot be inferred. In order to conclude with any certainty whether a desirable motivational climate is responsible for changing the quality of motivation of students, valuation of PE or even leisure time sport participation, intervention research is needed. Moreover, in this study the motivational PE climate was determined by the perceptions of students only. There is no data on which it can be concluded whether the PE classes were, in fact, mastery or performance orientated or psychological need supportive or frustrating. We removed 268 students from the sample due to missing data. It cannot be ruled out that these students were not representative of the whole group. However, we expect the possible impact of this (with a remaining N = 2562) to be within acceptable limits.

Furthermore, although this study provides a quantitative insight into students' perceptions of the motivational learning climate in Dutch secondary school PE, qualitative research is needed to gain more detailed insight and understanding of students' experiences and needs in PE.

## Conclusion

This study demonstrated that secondary school students in the Netherlands can be grouped in distinct motivational profiles based on how they perceive the PE environment. These profiles

can be of value to aid teachers' understanding of the differential affective experiences students can have during PE lessons. Our study may also provide the basis for future research aimed at the development of instructional strategies and design principles that take into account the motivational differences between students.

## Supporting information

**S1 File. Dataset profiling secondary school students.**
(XLSX)

## Author Contributions

**Conceptualization:** Gwen Weeldenburg, Lars B. Borghouts, Menno Slingerland, Steven Vos.

**Formal analysis:** Gwen Weeldenburg.

**Investigation:** Gwen Weeldenburg.

**Methodology:** Gwen Weeldenburg.

**Writing – original draft:** Gwen Weeldenburg.

**Writing – review & editing:** Lars B. Borghouts, Menno Slingerland, Steven Vos.

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
