## [Decision Letter · Decision Letter 0]

15 Jan 2020

PONE-D-19-32473

Similar but different: Profiling Secondary School Students based on their Perceived Motivational Climate and Psychological Need-Based Experiences in Physical Education

PLOS ONE

Dear Ms Weeldenburg,

Thank you for submitting your manuscript to PLOS ONE. After careful consideration, we feel that it has merit but does not fully meet PLOS ONE’s publication criteria as it currently stands. Therefore, we invite you to submit a revised version of the manuscript that addresses the points raised during the review process.

We would appreciate receiving your revised manuscript by Feb 29 2020 11:59PM. To enhance the reproducibility of your results, we recommend that if applicable you deposit your laboratory protocols in protocols.io, where a protocol can be assigned its own identifier (DOI) such that it can be cited independently in the future. For instructions see: http://journals.plos.org/plosone/s/submission-guidelines#loc-laboratory-protocols

We look forward to receiving your revised manuscript.

Kind regards,

Heather Erwin

Academic Editor

PLOS ONE

Additional Editor Comments (if provided):

I truly enjoyed reading this manuscript on profiling secondary students in physical education based on motivation and psychological needs. It was well-written and easy to read and follow. The methods appear to be sound, and the data analyses are solid. The discussion is well-organized and explains each of the research questions and elaborates on the results. While motivation in physical education is of very high interest to me, I am not an expert in the area. However, I do believe most of the appropriate literature has been referenced.

Lines 106 and 110 are incomplete sentences.

Is there any data to determine if the classes/teachers are, in fact, are mastery or performance climate? Or is this all based on the students’ perceptions? Please clarify.

As a field, we encourage physical education teachers to employ a mastery climate due to the seemingly negative impacts of a performance-related climate. Are the results of this study suggesting that a performance climate may be more beneficial for some students, depending on their profile? Please elaborate.

Reviewers' comments:

Reviewer's Responses to Questions

**Comments to the Author**

1. Is the manuscript technically sound, and do the data support the conclusions?

Reviewer #1: Yes

2. Has the statistical analysis been performed appropriately and rigorously? 

Reviewer #1: Yes

3. Have the authors made all data underlying the findings in their manuscript fully available?

Reviewer #1: Yes

4. Is the manuscript presented in an intelligible fashion and written in standard English?

Reviewer #1: Yes

5. Review Comments to the Author

Reviewer #1: Great introduction and reason to conduct the study. Good explanations of SDT and AGT. Very detailed methods section and results section. The graphs in the results section were very understandable and went along with the descriptions.

The one thing that I would state as a limitation would be the measurement for leisure time sport participation. Is regular physical activity considered sport participation in the Netherlands? I feel students might find it confusing and might not answer accurately on this type of questionnaire.

6. PLOS authors have the option to publish the peer review history of their article (what does this mean?). If published, this will include your full peer review and any attached files.

Reviewer #1: Yes: Kenneth Murfay

---

## [Author Response · Author response to Decision Letter 0]

22 Jan 2020

We thank you for giving us the opportunity to revise and resubmit our manuscript entitled ‘Similar but different: Profiling Secondary School Students based on their Perceived Motivational Climate and Psychological Need-Based Experiences in Physical Education’(PONE-D-19-32473). 

We are grateful to the reviewers for the positive evaluation and the clear and constructive feedback which enabled us to improve our manuscript. 

We thank you in advance for reconsidering our manuscript for publication.

Sincerely,

Gwen Weeldenburg (on behalf of all authors)

---

## [Editor Report · Decision Letter 1]

27 Jan 2020

Similar but different: Profiling Secondary School Students based on their Perceived Motivational Climate and Psychological Need-Based Experiences in Physical Education

PONE-D-19-32473R1

Dear Dr. Weeldenburg,

We are pleased to inform you that your manuscript has been judged scientifically suitable for publication and will be formally accepted for publication once it complies with all outstanding technical requirements.

With kind regards,

Heather Erwin

Academic Editor

PLOS ONE

Additional Editor Comments (optional):

Thank you for the addressing the comments. I am impressed with the quality of this paper. I am happy it will be a part of this journal.

---

## [Editor Report · Acceptance letter]

3 Feb 2020

PONE-D-19-32473R1 

Similar but different: Profiling Secondary School Students based on their Perceived Motivational Climate and Psychological Need-Based Experiences in Physical Education 

Dear Dr. Weeldenburg:

I am pleased to inform you that your manuscript has been deemed suitable for publication in PLOS ONE. Congratulations! Your manuscript is now with our production department. 

With kind regards,

on behalf of

Dr. Heather Erwin 

Academic Editor

PLOS ONE